# Prevalence of insomnia and associated factors among postpartum mothers in Southern Ethiopia, a community-based cross-sectional study

Mesfin Difer Tetema[1]*, Kassahun Fikadu[2], Gistane Ayele[3], Gudeta Beriso Jima[1], Berhanu Negesse Kebede[2], Awol Arega Yimer[2], Legese Fekede Abza[4], Mebratu Demissie[1], Kenzudin Assfa Mossa[5], Seid Jemal Mohammed[1], Ayele Sahile Abdo[1], Mangistu Abera[1]

1 Department of Midwifery, College of Medicine & health Sciences, Wolkite University, Wolkite, Ethiopia, 2 Department of Midwifery, College of Medicine & health sciences, Arba Minch University, Arba Minch, Ethiopia, 3 School of Public health, College of Medicine & health sciences, Arba Minch University, Arba Minch, Ethiopia, 4 Department of Nursing, College of Medicine & health sciences, Wolkite University, Wolkite, Ethiopia, 5 Department of Public health, College of medicine & health sciences, Wolkite University, Wolkite, Ethiopia

* mesfindifer@gmail.com

**Data Availability Statement:** All relevant data are within the paper and its Supporting Information files.

## Abstract

Insomnia has become a global public health concern, particularly among postpartum women. Minimal sleep interruption related to newborn care is normally expected, insomnia, however has negative impact during the postpartum period. Although its causes and contributing factors are poorly understood, it has a wide-ranging impact on the mother and her infant. So far, studies in Ethiopia have focused on the general community, neglecting mothers in the postpartum period. Thus, this study aimed to assess the prevalence of insomnia and the factors associated with it. A community-based cross-sectional study included 451 study participants who were chosen using a simple random sampling technique. A structured, pretested, and interviewer-administered questionnaire was used to collect data. After entering the data into EpiData version 3.1, it was exported to the Statistical Package for Social Sciences version 26 for analysis. Bivariable and multivariable binary logistic regression analyses were carried out. Variables with a P-value of 0.2 in the bivariable analysis were included in the multivariable analyses. Those with a P-value of 0.05 were declared statistically significant in the final model. The current study included 444 mothers in total. Insomnia was prevalent among 23.2% (95% CI: 19.3%, 27.4%) of mothers who had given birth within the previous 12 months. Insomnia was associated with unplanned index pregnancy [AOR = 4.4, 95% CI (2.2, 8.7)], alcohol consumption [AOR = 3.0, 95% CI (1.4, 6.5), low social support [AOR = 9.7, 95% CI (4.4, 21.1)], medium social support [AOR = 2.2, 95% CI (1.1, 4.3)] and depression [AOR = 10.7, 95% CI (5.7, 20.0). A planned index pregnancy, abstaining from alcohol, and recognizing and treating postpartum depression were all advised.

**Funding:** The author(s) received no specific funding for this work.

**Competing interests:** The authors have declared that no competing interests exist.

**Abbreviations:** AIS, Athens Insomnia Scale; AOR, Adjusted Odds Ratio; COR, Crude Odds Ratio; CI, Confidence interval; MSSS, Maternity Social Support Scale.

## Introduction

Insomnia is a condition in which a person reports not getting enough good sleep or having difficulty initiating and/or maintaining the sleep required to feel rested and rejuvenated when they wake up [1]. In most cases, insomnia is classified as transient (lasting a few days), short-term (lasting up to three weeks), or long-term (lasting more than three weeks) [2]. It has no apparent etiology; it might be the sole symptom or, more commonly, co-occur with other medical and psychological disturbances or be linked to neurobiological, hormonal, socio-demographic, and other co-morbidities [1, 3, 4]. Long-term health consequences of insomnia include decreased quality of life as well as physical and psychological illness [5]. Estimating the extent of insomnia is influenced by the population being researched as well as the criteria utilized to define it [1]. Women have a 40% higher risk of insomnia than men [3, 6]. The risk is even higher during the postpartum period [6, 7]. Mother-infant bonding, quality of life, maternal health, and the physical, emotional, and cognitive health of the child are all negatively impacted by insomnia during the postpartum period [8, 9]. During the postpartum period, properly recognizing, understanding the associated variables, and executing the vast variety of therapeutic measures for insomnia is regarded as quite significant [1, 6, 10]. So far, studies have targeted the general population, and women during the postpartum period have been overlooked. On the other hand, there is a scarcity of community-level data on the prevalence and contributing factors of insomnia in Ethiopia, particularly in the research area. As a result, the purpose of this study is to fill in the knowledge gap by determining the prevalence of insomnia and its associated factors among mothers who gave birth in the study region within the previous 12 months.

## Methods and materials

### Study design, period, and setting

A community-based cross-sectional study was done in Arba Minch Town, Southern Ethiopia, from April 19 to May 19, 2021. Arba Minch Town is the administrative center of the Gamo zone in southern Ethiopia. The town is 500 kilometers south of Ethiopia's capital, Addis Ababa, and 275 kilometers southwest of Hawassa, the capital of Ethiopia's Southern Nation Nationalities and People Region (SNNPR). There are eleven kebeles in the town, with a total population of 135,452 people, 49.7% of whom are female [11]. According to a report from the town's health department, 3765 moms gave birth in the town in the previous 12 months. There are also two hospitals, two health clinics, and around twenty private health services in the town.

### Population and eligibility criteria

All mothers who gave birth within the previous 12 months during the study period were the source population. Those mothers who were selected randomly during the data collection period were the study population. All mothers who gave birth within the previous 12 months during the study period were included in the study. Those mothers who were in their first six postpartum days were excluded from the study.

### Sample size determination

The minimum sample size was calculated using a single population proportion formula: By considering the estimated proportion of insomnia (P) of 21.8% taken from the study conducted in Bahir Dar, Ethiopia [12], a 95% confidence interval (CI), and a 4% marginal error (d) with a 10% contingency for non-response rate, the sample size yielded 451.

## Sampling procedure

A total of 451 study participants were drawn by a simple random sampling procedure using computer-generated random numbers. Health extension workers in the individual kebeles provided lists of homes with mothers who had given birth in the previous 12 months, which were utilized as a sample frame to design the research unit. Each kebeles received samples per their size. Using their house numbers and health extension workers as guides, randomly selected mothers were traced.

## Data collection tool and measurements

An interviewer-administered structured questionnaire was used to collect the data. The full-version (8-item) standardized Athens insomnia scale (AIS) was used to measure insomnia [13]. Those participants who scored 6 and above by using the AIS were considered to have insomnia [12]. Depression was measured using Edinburgh's postnatal depression scale (EPDS) [14]. It is a 10-item self-reporting scale with scores of 0-3 based on a 1-week recall and specifically designed to screen for postpartum depression. Those mothers who scored $\geq 13$ cut-off points using EPDS were considered to have depression [15]. Substance use (alcohol, khat, and cigarettes) was considered when mothers had used those substances at least once in the past 30 days [12]. Social support was measured using the Maternity Social Support Scale (MSSS) [16]. It was classified into three categories; high social support for those who scored 24–30, medium social support for those who scored 18–23, and low social support for those who scored below 18 using MSSS [17]. Four BSc nurses conducted data collection using face-to-face interviews.

## Data quality control

A standardized tool was used to collect the data to ensure data quality. The tool was pre-tested in Wolayita Sodo town two weeks before the real data collection period, with 5% of the sample. A unique identifying number was assigned to each questionnaire. Data collectors and supervisors received adequate training on the research objectives, data collection methods and techniques, and interviews. All the data collection processes were closely supervised, and the collected data were actively checked by the lead investigator and supervisor to verify that the information obtained was complete and consistent, and urgent corrective steps were taken as needed. Finally, the data were entered into EpiData version 3.1 software after being checked for completeness and properly coded with a unique identification number. Two separate data clerks did double data entry to cross-check in consistency of data entry. Simple frequencies and cross-tabulation were done to look for missing values and outliers. This was then cross-checked by reviewing hard copies of the collected data.

## Statistical analysis

The data were exported from EpiData to SPSS version 26 for analysis. Frequency, tables, figures, mean, and standard deviation (SD) were used to present the descriptive data. The association between the independent variables and the dependent variable was determined using a bivariable binary logistic regression analysis. Variables with a p-value of <0.2 in the bivariable binary logistic regression analysis were added to the multivariable binary logistic regression analysis to control the confounders. Variance inflation factor (VIF) >10, and Tolerance <0.1 were considered suggestive of multicollinearity; however, no multicollinearity was detected during the analysis. The model's fitness was assessed using the Hosmer and Lemeshow goodness-of-fit test. It was found to be insignificant (p-value = 0.729), indicating that the model

had been fitted. Adjusted Odds Ratio (AOR) with a 95% CI was estimated to show the strength of the association between the independent variables and the dependent variable after controlling for the effects of confounders. Independent variables with a P-value < 0.05 were declared to have a statistically significant association with the outcome variable.

### Ethical considerations

Initially, the idea was examined and authorized by the Arba Minch University, college of medicine and health sciences. An official letter was obtained from the Institutional Research Ethics Review Board of the college with a reference number IRB/1069/21, as well as authorization was obtained from the Arba Minch local Health Administration. During data gathering, data collectors and supervisors wore face masks and followed the idea of physical distancing as well as other precautions not to impose a risk of Covid-19 on study participants. Each study participant signed an informed consent form. The data collectors gave each participant explicit information about the study, including its goal, the importance of their participation in the study, and for them, being aware of their involvement had no compensation, but told them the study's results helped them indirectly. They were informed that the interview could take approximately ten minutes and that it could cause some minor discomfort. Aside from that, the mothers were not harmed by this study. After assuring them of their right to withdraw from participation at any point if they felt uncomfortable doing so without causing any consequence, they were asked to indicate their willingness to participate.

### Patient and public involvement

No patient involved.

## Results

### Socio-demographic characteristics of the respondents

A total of 444 respondents were participated in this study, making a response rate of 98.45%. The mean age (± SD) of the respondents was 30.55 (± 5.98). More than half of the respondents (55.6%) were Gamo, and a fifth (17.1%) were Wolayita by ethnicity. Moreover, two-fifths (43.9%) and 42.3 percent of the participants (42.3%) were protestant and orthodox Christians, respectively. Ninety-five percent (n = 422) of the study participants were married. About a quarter (24.3%) of the study participants had joined college. In terms of occupation, nearly a third of the respondents (n = 139) were housewives. The median monthly income of the respondents was 7000 Ethiopian Birr (ETB) with an inter-quartile range of 4517 ETB (Table 1).

### Obstetric characteristics of the respondent

Of the study participants, 349 (78.6%) were multiparous, whereas 95 (21.4%) were primiparous. Eighty-nine percent of the respondents had at least one ANC follow-up during their index pregnancy. Around 426 (95.9%) of the respondents gave their last birth at a health facility (Table 2).

### Clinical and behavioral characteristics of the respondent

Coffee consumption in the evening or at night was the most commonly (41.1%) used substance among the study participants. Of the total respondents, 84 (18.9%) of the mothers had depression. Among the participants, 26 (5.9%) had co-morbid illnesses. More than half (239)

**Table 1. Socio-demographic characteristics of mothers who gave birth within the previous 12 months in Southern Ethiopia, 2021 (n = 444).**

| Characteristics | Category | Frequency | Percent |
|---|---|---|---|
| Age of the respondents | Below 25 years | 79 | 17.8% |
| | 25-34 years | 259 | 58.3% |
| | 35 and above | 106 | 23.9% |
| Educational status of the mother | Can't read and write | 56 | 12.6% |
| | Can read and write | 76 | 17.1% |
| | Grade 1-8 | 77 | 17.3% |
| | Grade 9-12 | 127 | 28.6% |
| | College and above | 108 | 24.3% |
| Religion | Protestant | 195 | 43.9% |
| | Orthodox | 188 | 42.3% |
| | Muslim | 45 | 10.1% |
| | Others | 16 | 3.6% |
| Ethnicity | Gamo | 247 | 55.6% |
| | Gofa | 49 | 11.0% |
| | Wolayta | 76 | 17.1% |
| | Amhara | 26 | 5.9% |
| | Oromo | 31 | 7.0% |
| | Other | 15 | 3.4% |
| Marital status | Married | 422 | 95.0% |
| | Single/divorced/widowed | 22 | 5% |
| Occupation of the mother | Housewife | 139 | 31.3% |
| | Merchant | 77 | 17.3% |
| | Government employee | 110 | 24.8% |
| | Farmer | 9 | 2.0% |
| | Daily laborer | 54 | 12.2% |
| | Student | 41 | 9.2% |
| | Other | 14 | 3.2% |

(53.8%) of the mothers had high social support, 135 (30.4%) of them had medium social support, and 70 (15.8%) of them had low social support (**Fig 1**).

## Prevalence of insomnia

Of the study participants, 23.2% (n = 103), (95% CI, 19.3%, 27.4%) of mothers who gave birth in the last 12 months in the study met criteria for insomnia.

## Factors associated with insomnia

Based on the results of bivariable analysis at a p-value of < 0.2, the variables statistically associated with insomnia among mothers who gave birth within the last 12 months were the age of the respondents, status of index pregnancy, ANC follow-up during the index pregnancy, mode of delivery, alcohol consumption, coffee consumption, social support, and depression. Multivariable logistic regression was done by taking the variables that were statistically significant in bivariable logistic regression into account simultaneously. Then the back-ward conditional regression method was used. The variables that persisted to be significantly associated with insomnia at a p-value of < 0.05 in the final model were the Status of index pregnancy, alcohol consumption, social support, and depression. Insomnia was significantly associated with the status of their index pregnancy. The odds of insomnia was 4.4 times [AOR = 4.4 (2.2, 8.7)] higher among mothers who had an unplanned index pregnancy when compared to those who

**Table 2. Obstetric characteristics of mothers who gave birth within the previous 12 months in Southern Ethiopia, 2021 (n = 444).**

| Characteristics | Category | Frequency | Percentage |
|---|---|---|---|
| Parity | Primiparous | 95 | 21.4% |
| | Multiparous | 349 | 78.6% |
| Status of index pregnancy | Planned | 364 | 82% |
| | Unplanned | 80 | 18% |
| ANC follow up during index pregnancy | Yes | 395 | 89% |
| | No | 49 | 11% |
| Number of ANC received during index pregnancy | 1 | 27 | 6.8% |
| | 2 | 48 | 12.1% |
| | 3 | 126 | 31.9% |
| | 4 and above | 194 | 49.1% |
| Place of delivery | Institution | 426 | 95.9% |
| | Home | 18 | 4.1% |
| Mode of delivery | Vaginal delivery | 368 | 82.9% |
| | C/S | 76 | 17.1% |
| Complications after birth | Yes | 36 | 8.1% |
| | No | 408 | 91.9% |
| PNC (≥1 visit) | Yes | 25 | 5.6% |
| | No | 419 | 94.4% |
| Types of infant feeding | Exclusive breast milk | 201 | 45.3% |
| | Complementary | 173 | 39% |
| | Mixed feeding | 62 | 14% |
| | Other | 8 | 1.8% |
| History of infant loss | Yes | 20 | 4.5% |
| | No | 424 | 95.5% |

had a planned index pregnancy. In addition, this finding showed, insomnia to be significantly associated with alcohol consumption. The odds of insomnia among mothers who were consuming alcohol was 3.0 times [AOR = 3.0 (1.4, 6.5)] higher than those who were not consuming alcohol. This study also revealed maternal social support to be significantly associated with insomnia. The odds of insomnia was 2.2 times [AOR = 2.2 (1.1, 4.3)] higher among mothers with medium social support when compared to those with high social support. The odds of insomnia was 9.7 times [AOR = 9.7 (4.4, 21.2)] higher among mothers with low social support than those with high social support. Depression was also significantly associated with insomnia in this study. The odds of insomnia was 10.7 times [AOR = 10.7 (5.7, 20.0)] higher among mothers who had depression when compared to mothers who had no depression (**Table 3**).

## Discussion

The overall prevalence of insomnia among mothers who gave birth within the previous 12 months in the study area was 23.2%. This finding is lower than studies conducted in Korea (50.5%) [18], Nepal (50.2% [19], Taiwan (61.6%) [20], and Iran (35.4%) [21]. This difference might be due to the magnitude of alcohol consumption in Ethiopia being low when compared to those countries [22]. However, the finding of this study was higher than studies conducted in Israel (10%) [9] and Canada (16%) [23]. This difference might be due to socio-demographic variations. The finding of this study is in-line with a study conducted in Bahir Dar Ethiopia (21.8%) [12], these consistent findings might show that insomnia is a troublesome issue among postnatal mothers throughout the country. In this study, insomnia was statistically

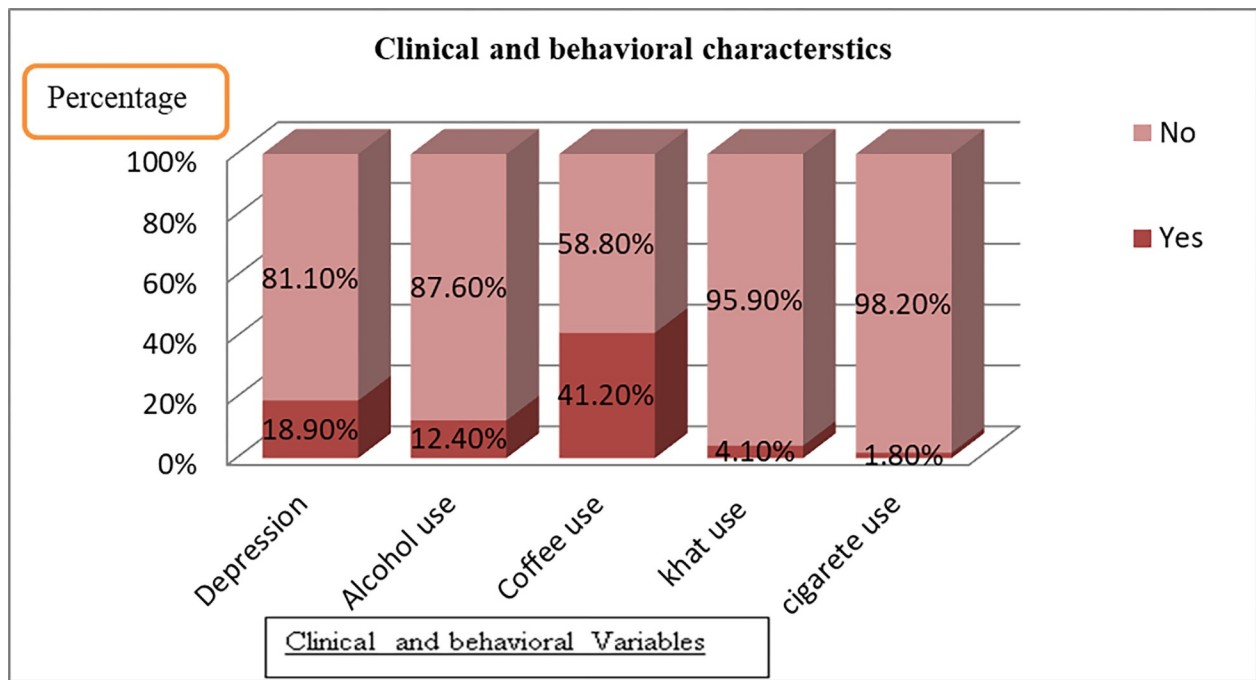

**Fig 1. Clinical and behavioral characteristics.**

**Table 3. Bivariable and multivariable logistic regression analysis depicting factors associated with insomnia among mothers who gave birth within the previous 12 months in Southern Ethiopia 2021(n = 444).**

| Characteristic | Category | Insomnia | | COR(95% CI) | AOR (95% CI) | p-value |
|---|---|---|---|---|---|---|
| | | Yes | No | | | |
| Age of the respondents | Below 25 years | 25 | 54 | 1.6(0.8, 3.1)* | 1.3(0.5, 3.3) | .573 |
| | 25-34 years | 54 | 205 | 0.9(0.5, 1.6) | 0.8(0.4, 31.6) | 0.468 |
| | 35 and above | 24 | 82 | 1 | 1 | |
| Status of index pregnancy | Unplanned | 45 | 35 | 6.8(4.0, 11.5)* | 4.4 (2.2, 8.7)** | .001 |
| | Planned | 58 | 306 | 1 | 1 | |
| ANC follow up | No | 20 | 29 | 2.6(1.4, 4.8)* | 1.9(0.8, 4.98) | .167 |
| | Yes | 83 | 312 | 1 | 1 | |
| Mode of delivery | C/S | 26 | 50 | 1.96(1.1, 3.4)* | 2.0(0.9, 4.4) | .082 |
| | Vaginal | 77 | 291 | 1 | 1 | |
| Alcohol use | Yes | 23 | 32 | 2.8(1.5, 5)* | 3.0 (1.4, 6.5)** | .005 |
| | No | 80 | 309 | 1 | 1 | |
| Coffee use | Yes | 54 | 129 | 1.8(1.2, 2.8)* | 1.3(0.7, 2.4) | .444 |
| | No | 49 | 212 | 1 | 1 | |
| Social support | Low | 49 | 21 | 19.97(10.3,38.6)* | 9.7(4.5, 21.2)** | .001 |
| | Medium | 29 | 106 | 2.3 (1.3, 4.2)* | 2.2(1.1, 4.3)** | .026 |
| | High | 25 | 214 | 1 | 1 | |
| Depression | Yes | 55 | 29 | 12.3(7.2, 21.2)* | 10.7 (5.7, 20.0)** | .001 |
| | No | 48 | 312 | 1 | 1 | |

*Significant at p <0.2

**significant at p < 0.05, 1= reference

associated with the status of the index pregnancy. The odds of insomnia among mothers who had an unplanned index pregnancy was 4.4 times higher when compared to those who had a planned index pregnancy. However, this finding is not supported by the study conducted in Bahir Dar Ethiopia [12]. This might be due to this study being community-based where many women with unplanned pregnancies fail to visit health facilities when compared to the study in Bahir Dar Ethiopia that was institution-based [12]. Alcohol consumption was also significantly associated with insomnia in this study; the odds of insomnia were 3 times higher among mothers who were consuming alcohol than those who were not consuming alcohol. This finding is also supported by studies done at Bahir Dar, Ethiopia [12]. This is due to alcohol, which is one of the most addictive psychoactive substances that can lead to physical and psychological dependence damaging emotional effects resulting in disturbed sleep patterns [24]. In this study, maternal social support was similarly linked to insomnia; the odds of insomnia was 2.2 times higher among mothers with medium social support when compared to those with high social support. In addition to this, the odds of insomnia were 9.7 times higher among mothers with low social support than those with high social support. These findings were also supported by a study conducted in Bahir Dar, Ethiopia [12]. This might be due to good social support may influence sleep patterns by instilling a sense of belonging and connectedness, inducing a positive mood state, and promoting positive health behaviors such as healthy sleep habits [25]. Based on the findings of this study, the odds of insomnia were 10.7 times higher among mothers who had depression when compared to those who had no depression. The finding was also supported by studies conducted in Taiwan [26], Norway [27] Turkey [28], Nepal [29], and Ethiopia [12]. This finding may be explained by the well established and bidirectional relationship between insomnia and depression [30].

## Conclusion

Insomnia during the postpartum period is a serious issue having negative consequence on the normal functioning of the mother and the health of the newborn. This study assessed the prevalence of insomnia and its associated factors among mothers who had given birth within the previous 12 months in the study area. A quarter of women experienced insomnia during the postpartum period in Southern Ethiopia. Unplanned index pregnancy, alcohol consumption, low to medium social support and depression were factors significantly associated with insomnia. Given the findings of the study, all concerned bodies were advised to learn about and act on the identified factors to promote maternal mental health as well as child's health and development. Therefore, the following recommendations were made for the stakeholders.

We wish to recommend that mothers have a planned pregnancy though utilization of family planning methods. We also wish to recommend mothers avoid alcohol consumption. These can help them by decreasing physical and psychological dependence as a result this helps them to adequately cope with the emotional changes during the postpartum period and promotes a healthy sleep pattern. As the Health extension workers (HEWs) and other health professionals are the primary caregivers, we wish to recommend them to create awareness for mothers on the adverse impacts of alcohol consumption. These can improve the mother's knowledge and attitude towards alcohol consumption during the postpartum period as a result mothers might acquire enough and good quality sleep. We wish also to recommend that HEWs and other health professionals devote their time to promoting family planning utilization as a means to avoid unplanned pregnancies. We also recommend that health professionals screen and effectively manage postpartum depression. Therefore, health intervention planners and providers should take these behavioral, social, and health-care-related factors into account while developing a post-partum insomnia prevention intervention.

## Strength and limitations of the study

Many of the maternal health problems are anchored in the community, hence the study focused on mothers throughout the postpartum period at the community level. The study's external validity was increased by including all kebeles in the study area. The study also relied on primary data. As a result, the data has a high level of trustworthiness. Because the instrument's reliability, validity, and sensitivity are high, the study enables the comparison of results. However, there are certain limitations to this research. In this study, moms were asked to estimate their sleep difficulty if it had occurred at least three times per week. As a result, recall bias could be present in this study. It is also defficult to estabilish the cause-effect relationship between some of the explanatory variables and the outcome variable. Comparison and discussion are challenging due to limited previous studies conducted in Ethiopia.

## Supporting information

**S1 File.**
(SAV)

## Acknowledgments

We would like to acknowledge Arba Minch University, College of Medicine and Health Sciences for the ethical approval. We would also like to express our gratitude to the study participants, data collectors, supervisors, and the Arba Minch town health department.

## Author Contributions

**Conceptualization:** Mesfin Difer Tetema, Kassahun Fikadu, Gistane Ayele, Mangistu Abera.

**Data curation:** Gudeta Beriso Jima, Berhanu Negesse Kebede, Awol Arega Yimer.

**Formal analysis:** Mebratu Demissie, Kenzudin Assfa Mossa, Ayele Sahile Abdo, Mangistu Abera.

**Funding acquisition:** Awol Arega Yimer.

**Methodology:** Mesfin Difer Tetema, Kassahun Fikadu, Gistane Ayele, Seid Jemal Mohammed, Mangistu Abera.

**Resources:** Berhanu Negesse Kebede.

**Software:** Kassahun Fikadu, Gistane Ayele.

**Supervision:** Mesfin Difer Tetema, Mebratu Demissie.

**Validation:** Kenzudin Assfa Mossa.

**Visualization:** Awol Arega Yimer.

**Writing – original draft:** Gudeta Beriso Jima, Legese Fekede Abza.

**Writing – review & editing:** Gudeta Beriso Jima, Ayele Sahile Abdo.

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
