## [Decision Letter · Decision Letter 0]

26 Mar 2024

PONE-D-23-29005Prevalence Of Insomnia And Associated Factors Among Postpartum Mothers in Southern Ethiopia, a community-based cross-sectional study.PLOS ONE

Dear Dr. Tetema,

Thank you for submitting your manuscript to PLOS ONE. After careful consideration, we feel that it has merit but does not fully meet PLOS ONE’s publication criteria as it currently stands. Therefore, we invite you to submit a revised version of the manuscript that addresses the points raised during the review process.

We look forward to receiving your revised manuscript.

Kind regards,

Anteneh Mengist Dessie, MPH

Academic Editor

PLOS ONE

2. In the online submission form, you indicated that [The data underlying the results presented in the study are available from the corresponding author]. 

Reviewers' comments:

Reviewer's Responses to Questions

**Comments to the Author**

1. Is the manuscript technically sound, and do the data support the conclusions?

Reviewer #1: Yes

Reviewer #2: Yes

2. Has the statistical analysis been performed appropriately and rigorously? 

Reviewer #1: Yes

Reviewer #2: Yes

3. Have the authors made all data underlying the findings in their manuscript fully available?

Reviewer #1: No

Reviewer #2: Yes

4. Is the manuscript presented in an intelligible fashion and written in standard English?

Reviewer #1: Yes

Reviewer #2: Yes

5. Review Comments to the Author

Reviewer #1: - In the abstract, consider one sentence distinguishing insomnia in postpartum from sleep disruption in postpartum due to direct childcare.

- I am not sure about the final sentence of the abstract. It feels strange to me to strongly advise a planned index pregnancy. On what grounds is this recommendation made? Consider amending this language. Correlation does not equate causation.

- P.7 the formatting and capitalization of academic affiliations is not consistent (ex. 1 Department of Midwifery, 2Department of midwifery).

- Introduction- I would consider a different definition of insomnia. The important distinction between postpartum insomnia and postpartum sleep disruption is- is the person having difficulty sleeping, outside of direct childcare. Basically, when given the opportunity, still struggling with their sleep.

- Consider removing this sentence or clarifying what is meant by ‘atypical’- Women have a 40% higher risk and an atypical presentation of insomnia than men (3, 6).

- I like the description of Arba Minch Town.

- P. 14 might be nice to include monthly income converted to USD or another currency so more readers can comprehend.

- P. 18- consider ‘met criteria for insomnia’ rather than ‘had insomnia’

- P. 21- I like the discussion opening, situating the findings in the context of literature from other countries

- P. 22 “This might be due to the definite link between depression and insomnia, as well as insomnia, being a manifestation of depression(30).” Consider amending the language to acknowledge the bi directional relationship for example “This finding may be explained by the well established and bi directional relationship between insomnia and depression.”

- For the conclusion- consider mentioning established implications of insomnia in postpartum.

- Please consider amending this sentence, it is hard to understand “As a result, the sustainable development goals for mental health can be achieved; even if it appears that world leaders are now off-track in achieving mental health targets, and hence, the following recommendations were made for actions to be performed by the stakeholders”.

Reviewer #2: Thank-you for this submission. I was curious about your definition of chronic insomnia - you mention difficulty sleeping that lasts for greater than 3 weeks. The definition I am more familiar with indicates chronicity begins after 3 months - was this a different definition specifically for post-partum patients?

I was impressed with your response rate and that you did a trial run for two weeks to ensure that the data collection was high quality.

6. PLOS authors have the option to publish the peer review history of their article (what does this mean?). If published, this will include your full peer review and any attached files.

Reviewer #1: No

Reviewer #2: No

---

## [Author Response · Author response to Decision Letter 0]

24 Apr 2024

Reviewer 1

Dear reviewer, thank you for your valuable comments and suggestions. It was very helpful. We have tried to address all the comments and suggestions on the main manuscript. Here, we have to clarify some important issues.

Thank you again for your time. 

Comment 

In the abstract, consider one sentence distinguishing insomnia in postpartum from sleep disruption in postpartum due to direct childcare.

Response: Thank you for the comment. We have amended it in the manuscript. 

I am not sure about the final sentence of the abstract. It feels strange to me to strongly advise a planned index pregnancy. On what grounds is this recommendation made? Consider amending this language. Correlation does not equate causation.

Response: Thank you for the comment. We have amended it in the manuscript. 

P.7 the formatting and capitalization of academic affiliations is not consistent (ex. 1 Department of Midwifery, 2Department of midwifery).

Response: Thank you for the comment. We have amended it in the manuscript.

- Introduction- I would consider a different definition of insomnia. The important distinction between postpartum insomnia and postpartum sleep disruption is- is the person having difficulty sleeping, outside of direct childcare. Basically, when given the opportunity, still struggling with their sleep..

Response: Thank you for the comment. The definition of insomnia is the same during the postpartum period when mothers are obliged for childcare with those having no such obligation. The tool (Athens Insomnia Scale) that we have used to measure insomnia during the postpartum period is also used for the general population. The issue is that women have added burden of insomnia due to many reasons including childcare during postpartum period.

- Consider removing this sentence or clarifying what is meant by ‘atypical’- Women have a 40% higher risk and an atypical presentation of insomnia than men (3, 6).

Response: Thank you for the comment. We have amended it in the manuscript.

- P. 14 might be nice to include monthly income converted to USD or another currency so more readers can comprehend.

Response: We appreciate your comment, the value of the local currency (Ethiopian birr (ETB)) when changed to USD or other currencies varies within a short period of time. As a result the value of ETB during the time of data collection would be difficult to gather. 

- P. 18- consider ‘met criteria for insomnia’ rather than ‘had insomnia’

Response: Thank you for the comment. We have amended it in the manuscript.

- P. 22 “This might be due to the definite link between depression and insomnia, as well as insomnia, being a manifestation of depression (30).” Consider amending the language to acknowledge the bi directional relationship for example “This finding may be explained by the well-established and bi directional relationship between insomnia and depression.

Response: Thank you for the comment. We have amended it in the manuscript.

- For the conclusion- consider mentioning established implications of insomnia in postpartum

Response: Thank you for the comment. We have amended it in the manuscript.

- Please consider amending this sentence, it is hard to understand “As a result, the sustainable development goals for mental health can be achieved; even if it appears that world leaders are now off-track in achieving mental health targets, and hence, the following recommendations were made for actions to be performed by the stakeholders”

Response: Thank you for the comment. We have amended it in the manuscript

Reviewer 2

Dear reviewer, thank you for your valuable comments and suggestions. It was very helpful. We really appreciate it. Here, we have to clarify the issue.

Comment: I was curious about your definition of chronic insomnia - you mention difficulty sleeping that lasts for greater than 3 weeks. The definition I am more familiar with indicates chronicity begins after 3 months - was this a different definition specifically for post-partum patients?

Response: Thank you for the comment. What we have written is not the definition of chronic insomnia. It is the definition of long-term insomnia. We have classified insomnia into transient, short-term and long-term (not acute and chronic). To our knowledge too chronic insomnia is insomnia lasting more than 3 months.

---

## [Decision Letter · Decision Letter 1]

1 Jul 2024

Prevalence Of Insomnia And Associated Factors Among Postpartum Mothers in Southern Ethiopia, a community-based cross-sectional study.

PONE-D-23-29005R1

Dear Dr. Tetema,

We’re pleased to inform you that your manuscript has been judged scientifically suitable for publication and will be formally accepted for publication once it meets all outstanding technical requirements.

Kind regards,

Anteneh Mengist Dessie, MPH

Academic Editor

PLOS ONE

Additional Editor Comments (optional):

Reviewers' comments:

Reviewer's Responses to Questions

**Comments to the Author**

1. If the authors have adequately addressed your comments raised in a previous round of review and you feel that this manuscript is now acceptable for publication, you may indicate that here to bypass the “Comments to the Author” section, enter your conflict of interest statement in the “Confidential to Editor” section, and submit your "Accept" recommendation.

Reviewer #2: All comments have been addressed

2. Is the manuscript technically sound, and do the data support the conclusions?

Reviewer #2: Yes

3. Has the statistical analysis been performed appropriately and rigorously? 

Reviewer #2: I Don't Know

4. Have the authors made all data underlying the findings in their manuscript fully available?

Reviewer #2: Yes

5. Is the manuscript presented in an intelligible fashion and written in standard English?

Reviewer #2: Yes

6. Review Comments to the Author

Reviewer #2: Thank-you for your responses and for addressing the concerns that were raised. We appreciate your recommendation that health care workers ask their patients about their sleep.

7. PLOS authors have the option to publish the peer review history of their article (what does this mean?). If published, this will include your full peer review and any attached files.

Reviewer #2: No

---

## [Editor Report · Acceptance letter]

12 Jul 2024

PONE-D-23-29005R1 

PLOS ONE

Dear Dr. Tetema, 

I'm pleased to inform you that your manuscript has been deemed suitable for publication in PLOS ONE. Congratulations! Your manuscript is now being handed over to our production team.

Kind regards, 

on behalf of

Mr. Anteneh Mengist Dessie 

Academic Editor

PLOS ONE